# Hazardous Waste Management of Buffing Dust Collagen

**DOI:** 10.3390/ma13071498

**Published:** 2020-03-25

**Authors:** Miroslawa Prochon, Anna Marzec, Oleksandra Dzeikala

**Affiliations:** Institute of Polymer and Dye Technology, Faculty of Chemistry, Lodz University of Technology, Stefanowskiego 12/16, 90-924 Lodz, Poland; anna.marzec@p.lodz.pl (A.M.); dzeikala.sandra@gmail.com (O.D.)

**Keywords:** buffing dust collagen (BDC), styrene–butadiene rubber (SBR), polymer composites, biodegradation

## Abstract

Buffing Dust Collagen (BDC) is a hazardous waste product of chromium tanning bovine hides. The aim of this study was to investigate whether BDC has the desirable properties required of modern fillers. The microstructural properties of BDC were characterized by elemental analysis (N, Cr_2_O_3_) of dry residue and scanning electron microscopy (SEM). The BDC was applied (5 to 30 parts by weight) to styrene butadiene rubber (SBR), obtaining SBR-BDC composites. The physicochemical properties of the SBR-BDC composites were examined by Fourier transform infrared analysis, SEM, UV–Vis spectroscopy, swelling tests, mechanical tests, thermogravimetric analysis (TGA), and differential scanning calorimetry (DSC). The biodegradability of the SBR-BDC composites and their thermo-oxidative aging were also investigated. The filler contributed to increase the cross-link density in the elastomer structure, as evidenced by enhanced mechanical strength. The introduction of a filler into the elastomer structure resulted in an increase in the efficiency of polymer bonding, which was manifested by more favorable rheological and mechanical parameters. It also influenced the formation of stable interfacial bonds between the individual components in the polymer matrix, which in turn reduced the release of compact chromium in the BDC filler. This was shown by the absorption bands for polar groups in the infrared analysis and by imaging of the vulcanization process.

## 1. Introduction

The leather industry is one of the most polluting sectors of the economy [1]. Wastes from the leather industry represent 5% by weight of the raw materials [2]. The processing of leather, in particular tanning, also involves toxic chemicals, which can escape into the environment. The most widely used tanning method is chromium tanning (85–90% of world production), which is a type of chemical modification [3,4,5]. It uses chromium (III) salts in the form of chromium (III) sulfate, in combination with sodium sulfate. The tanning process imparts the skin with desirable properties, such as strength, thinness, and hydrothermal resistance [3,6], while also simplifying further processing. To ensure the durability and elasticity of the skin after tanning, it is subjected to a buffing process. A by-product of that process is buffing dust collagen (BDC). Buffing dust collagen is formed in the process of fat liquoring leather. According to the literature [6], chromium can attach to the active sites of collagen. Molecular modeling and IR analysis confirm that chromium can react with amino as well as carboxylate groups. Each ton of raw material generates ~0.6% buffing dust. If this type of waste goes to landfill, it may be hazardous for the environment, because the oxidation state of chromium salts changes from toxic III to VI, resulting in dangerous chromium salt products [4,7]. Another way to deal with such waste is incineration. Leather wastes have high calorific value (12–14 MJ/kg). However, sulfides are formed during their combustion, Volatile Organic Compounds (VOC) and greenhouse gases may be emitted to the atmosphere [5,8,9]. A final way of dealing with BDC waste is collagen extraction [2,10,11]. Due to the environmental impact of buffing dust and chrome shavings, tanning waste has become the object of intense research.

Studies have shown that BDC and chrome shavings collagen (CSC) can be successfully used as fillers in rubbers [12,13,14,15,16,17]. The dust form should simplify its dispersion in an elastomeric matrix [11]. Buffing dust can be introduced into natural latex rubbers and used as a filler in dust systems. Chromium strings have been applied in acrylonitrile butadiene (NBR) and styrene butadiene (SBR) rubber matrices [18,19]. Kowalska et al. [20] subjected leather waste from pork skins to alkaline reagents, which increased their polymer bonding efficiency and resulted in improved stabilization of interfacial interactions, thereby reducing the evolution of chromium. This led to improved mechanical parameters, which in turn increased the collagen added to PVC (polyvinyl chloride)-produced materials. Adding particles of buffing dust to poly(vinyl chloride) increased its Young modulus, the value of melt flow index (MFI), and susceptibility to biodegradation. Chrońska-Olszewska and Przepiórkowska [13] report that when applied in the form of a leather shavings/dust mixture to NBR and *X*NBR, the filler produced biodegradable collagen–elastomer materials with improved mechanical properties and hardness. The leather shavings/dust mixture was an active filler for NBR and *X*NBR.

Residues after tanning are not of particular value, and the Cr (III) salts may naturally turn into toxic Cr (VI) waste. Zhou et al. [21] used chromium-heated leather with active zirconium particles as a material for removing fluoride ions from groundwater. Research is also being carried out to transform tanning waste into carbon adsorbents at low temperatures, below 600 °C, e.g., using ZnCl_2_ as the activating agent [22]. Leather waste has been used as a filling and stabilizing additive for bituminous and asphalt masses (Stone Matrix Asphalt (SMA)), improving their mechanical parameters, creep resistance, hardness, and humidity [23]. Ma et al. [24] developed a mesoporous material by high-temperature carbonization of chrome-tanned leather waste, which was then used for the electrodes in supercapacitors. The material was characterized by a high specific surface, low resistivity, and a high concentration of functional groups containing oxygen and nitrogen atoms. The admixture of leather waste with gravity substitutes for natural rubbers, acrylonitrile butadiene, or polyvinyl alcohol has led to interesting results [25,26]. The creation of hydrogen bonds is promoted, as well as chelation at interfaces, for example, between PVA and leather shavings, leading to greater compatibility of the tested centers. The elastomers of the tested rubbers also showed a significant increase in tear strength, due to the influence of skin particles.

The aim of the present study was to research the effect of applying buffing dust as a filler to styrene–butadiene rubber. There have been no previous reports of using BDC sanding dust from chrome tanning processes as a filler for SBR rubber in combination with a conventional seeding unit. Before being introduced into the elastomer matrix, the BDC was characterized by FTIR, SEM analysis, elemental analysis (including determination of chromium content), dynamic light scattering (DLS), and the DBP test. The crushed dust collagen reacted with the elastomer matrix and other components in the mixtures, as was confirmed by FTIR and mechanical studies.

## 2. Materials and Methods

### 2.1. Materials

#### 2.1.1. Rubber and Other Ingredients

The continuous phase was Styrene–Butadiene Rubber (SBR), KER 1500, from Bayer AG Company, Leverkusen, Germany. The other components were zinc oxide pure (ZnO) from LANXESS Deutschland GmbH, Augsburg, Germany; sulfur pure (S_8_) (density 2.07 g/cm^3^) from Siarkopol Tarnobrzeg Sp. z o.o., Tarnobrzeg, Poland; technical stearin, from Torimex Chemicals Ltd. Sp. z o.o., Konstantynów Łódzki, Poland; MBTS pure from Accelerator Bayer AG Company, Leverkusen, Germany; and toluene from Chempur Company, Piekary §láskie, Poland.

#### 2.1.2. Buffing Dust Collagen (BDC)

Buffing dust was sourced as a waste product from various batches of cattle skins from Kalisz Tabbery, Kalskór S.A., Kalisz, Poland. The black dust (i.e., basic chromium sulfate Cr(OH)SO_4_, chromium salts, vegetable tannins, etc.) was generated as a result of grinding tanned leather in the final stage of leather production, after dyeing and greasing, in a chromium system. The BDC was mixed, sieved through 2 mm sieves on a vibrating screen (AS200 Control, Retsch GmbH, Haan, Germany), and then conditioned at 50 °C for 5 h in a Binder thermal chamber (Binder GmbH, Tuttlingen, Germany). The content of chromium (III) as Cr_2_O_3_ varied from 4.25% to 4.48%, in accordance with the PN-EN standard ISO 4684: 2006 (U) [13,14]. The BDC fiber had a diameter of 0.2 mm.

### 2.2. Preparation of Composites

After the preparation step (see Section 2.1.2), the BDC fillers were applied to the styrene butadiene rubber at different concentrations. The rubber mixtures were prepared using a mixing mill (Bridge type milling machine, London, UK) with a roll temperature of 27–37 °C and friction of 1.1. The parameters of the rolling mill are as follows; roller length: L = 450 mm; roll diameter: D = 200 mm; rotational speed of the front roller: Vp = 20 rpm; width of the gap between the rollers: 1.5–3 mm. The mixtures were prepared for 6 min, then packed in foil and stored at 2–6 °C. The compositions of the rubber composites are given in Table 1. The tests were carried out at room temperature under normal pressure.

## 3. Research Techniques

The curing kinetics of the SBR compounds were studied in a moving die rheometer (MonTech MDR 300, Buchen, Germany) at 150 °C, according to the ISO 3417 standard. The rubber compounds were vulcanized according to the optimal curing time (τ_90_) in a standard electrically heated hydraulic press at a temperature of 150 °C, with a pressure of 15 MPa. The mechanical properties of the prepared composites were tested using a Zwick universal testing machine, model 1435 (according to PN-ISO 37:1998). The tensile strength (TS_b_) and percentage elongation at break (E_b_) were determined. Hardness testing (H, °Sh) was carried out using a Shore electronic hardness tester, type A, with a force of 12.5 N, according to standard PN-80C-04238 (Zwick/Roell, Herefordshire, UK).

The polymer–solvent interaction parameter (0.378 for SBR rubber in toluene solvent) was determined based on the equilibrium swelling method (according to PN-ISO 1817:2001/ap1:2002). The cross-link density (ν, 10^4^ mol/dm^3^) was calculated as the volume fraction of rubber in the swollen material, and V_S_ = 106.3 mol/cm^3^ for the molar volume of solvent (toluene) [13,14,15,16,17,27].

Cross-linking density, ν, was calculated on the basis of Flory–Rehner’s Equation (1):(1)$$v=\frac{\left[\ln\left(1-V_{r}\right)+V_{r}+{\mu V}_{r}^{3}\right]}{V_{O}\left(V_{r}^{\frac{1}{3}}-\frac{V_{r}}{2}\right)}$$
where µ is the Huggins parameter for the uncross-linked polymer–solvent system and V_r_ is the molar volume of the swelling solvent.

The thermo-oxidative experiments were performed in a convection oven and a thermal chamber with air circulation (Binder GmbH, Tuttlingen, Germany). An unstressed sample was exposed to the action of circulating air at 70 °C for 168 h. To study the deformation energy of the vulcanizates as a result of biological aging, the aging factor (S) was calculated according to Equation (2):(2)$$S=\frac{{\mathrm{TS}}_{b1}\times E_{b1}}{{\mathrm{TS}}_{b2}\times E_{b2}}$$
where ${\mathrm{TS}}_{b1}\times E_{b1}$ is the tensile strength (MPa) and elongation at break (%) after thermal-oxidative aging or soil test and ${\mathrm{TS}}_{b2}\times E_{b2}$ is the tensile strength (MPa) and elongation at break (%) before thermal-oxidative aging or the Soil Test.

A biodecomposition test was performed in soil with paddle-shaped samples with dimensions of 7.5 cm by 1.25 cm, and sampling of 0.4 cm. The samples were placed in an active universal soil (10 cm depth) and incubated at a temperature of 30 °C with 80% RH for 90 days in a climatic chamber (HPP 108 Memmert GmbH, Schwabach, Germany). Tests were carried out according to PN-EN ISO 846. The soil test was analyzed following the method described by Tadeusiak et al. [15]. Surface topography of the composites was conducted after the soil tests, using photos taken with a Canon CanoScan 4400F device. The morphology of the BDC powder and SBR composites were analyzed using a scanning electron microscope (SEM), Zeiss Ultra Plus (Bruker). Prior to the analysis, the samples were coated with a carbon target using a Cressington 208 HR system [13,16]. A Nicolet 6700 FT-IR spectroscope (Thermo Scientific, Waltham, MA, USA) with Fourier transformation and ATR snap was used to determine the characteristics of the composites. Analysis performed in the range of 4000 to 400 cm^−1^ [13]. Differential scanning calorimetry (DSC) was performed using a DSC1 analyzer (Mettler Toledo, Netzsch, Switzerland) at a heating rate of 10 °C/min. The SBR samples were heated from −150 °C to 350 °C under a nitrogen atmosphere. Thermogravimetric analysis (TGA) was performed using a TGA/DSC1 analyzer (Mettler Toledo, Netzsch, Switzerland). The heating rate was 10 °C/min under a nitrogen atmosphere, across a temperature range of 25 to 900 °C. DSC was analyzed as described by Prochoń et al. [28]. The changes in color of the samples after the thermo-oxidative aging process and soil test were studied using a UV–Vis CM-3600d spectrophotometer (Konica Minolta, London, UK). The difference in color was expressed as the color change parameter dE × Lab, where L is the level of lightness or darkness, a is the relationship between redness and greenness, and b is the relationship between blueness and yellowness [13].

Elemental analyses of the carbon, hydrogen, and nitrogen elements in the BDC powder were carried out using a Vario EL III analyzer equipped with special adsorption columns and a thermal conductivity detector (TCD). The absorption of dibuthylphtalate (DBP) by the BDC powder was measured using an Absorptometer C (Brabender mixer, Brabender GmbH & Co. KG, Duisburg, Germany). The sizes of the BDC particles were determined in water (filler concentration of 0.01 g/250 mL) using the dynamic light scattering (DLS) method on a Zetasizer Nano (Malvern Instruments, Malvern, Great Britain) analyzer [14].

## 4. Results and Discussion

### 4.1. Characterization of BDC Powder

The valence vibrations of bands derived from methyl and methylene groups in the tested BDC dust are visible in the wave number range of 3420 to 3479 cm^−1^ (Figure 1). Maxima of the bands appeared in the dust spectrum at a lower wave number intensity. Bands at 1750 cm^−1^ (–COO^−^) indicate the presence of fatty substances [13,14,28].

In the area from 3200 to 3500 cm^−1^ there is a wide band corresponding to the valence vibrations ofhydroxyl (–OH) side chains and end groups, with higher intensity in the dust spectrum. In the range of 1650 to 1500 cm^−1^, a band of deformation vibrations of the first-order amide appears for (C=O) and second-order amide (NH). The effect of chromium on interactions with other components of the BDC powder, and thus other interactions, was visible through shifts in the absorption bands, reducing their intensity, etc. The characteristic absorption band at 1654 cm^−1^ may be attributed to the possible mechanism of interaction of Cr with the protein-like system –Cr–OOC– (Figure 1b) [6]. The band of COOH stretching vibrations derived from amide I is shifted from 1660 cm^−1^. There is also a characteristic wide absorption band for chromate samples at about 1000 cm^−1^, which formed as a possible result of the interaction of chromium with a carboxyl group. The presence of Cr–O–Cr bonds is indicated by the bands between 510 and 650 cm^−1^ [6,13].

The shape, particle size, and specific surface of the filler are known to have a decisive impact on the strength of rubber–filler joints. Dust morphology was assessed based on photos taken using SEM, as shown in Figure 2. Dust agglomerates are visible as primary particles with a regular structure: elongated, insulated fibers with a wide size distribution from several hundred nanometers to several micrometers.

One collagen macrofibrillary fiber is connected to several or even several dozen individual helical microfibers (Figure 2a), with diameters of ~3–4 micrometers and longitudinal segmentation.

The particle size distribution of the tested BDC was measured by dynamic light scattering (DLS) in an aqueous solution. The particle size distribution, which is a compilation of measuring the length and diameter of particles oriented in the laser light field, was in the range from 469 to 295 nm. The isoelectric Point (IEP) was at pH 5.9. There was an appropriately small area of 9 m^2^/g. However, based on the elemental analysis of BDC, the nitrogen conversion to protein substance, which determines the nitrogen content in the collagen, was 7.92%. The Cr converted to Cr_2_O_3_ was at 4.48%. Dry matter was 89.49%, and ash was ~7% [13,14].

Oil number is one of the important factors that measure the structure and surface of fillers. From a morphological point of view, fillers have the ability to form aggregates or agglomerates. To prevent the influence of physico-chemical interactions, fillers are often subjected to modifications aimed at changing their structural or surface characteristics. The structure of the BDC had been changed under the influence of the chemical modification processes during tanning.

As can be seen in the Annex (see Appendix A
Appendix A), after the addition of dibutyl phthalate (DEP), the BDC molecules begin to approach each other and form agglomerates. There is increasing resistance to mixing, due to the higher torque. At the moment of maximum saturation with PBT, the process ends, obtaining maximum torque. The maximum torque was 248.4 mNm. The filling volume efficiency was determined using the Medalia model, according to Equation (3):(3)$$\varphi_{\mathrm{eff}}=0.5\emptyset \left[1+\left(1+\frac{0.0213\left(\mathrm{DBP}\right)}{1.46}\right)\right]$$
where $\varphi_{\mathrm{eff}}$ is the actual volume of the filler. The efficiency of the BDC was 4.33 mL/g. The moisture content of the BDC particles was in the range of 10^2^ to 10^4^ nm. This important parameter classifies the BDC filler in the group of semi-reinforcing fillers. The filler shows a high degree of orderliness and a tendency to form aggregates.

### 4.2. Characterization of SBR/BDC Composites

#### 4.2.1. Rheometric Properties

The BDC filler clearly affected the curing characteristics of the SBR mixtures. The kinetic parameters of the BDC-based compounds differed significantly from those of the unfilled compound, as shown in Table 2. The incorporation of BDC into the SBR compound resulted in higher viscosity and stiffness, as reflected in the torque values (LH, ∆L), which are much higher than those of the SBR samples. The incorporation of the biofiller resulted in higher cross-link density, as the ∆L parameter is an indirect measurement of the degree of elastomer cross-linking. The unfilled sample exhibited shorter time of vulcanization and scorch time than the filled composites. The scorch and vulcanization times increased with increasing concentrations of BDC, showing the highest values for 30 phr concentration (τ_02_ = 3.5 and τ_90_ = 38.5). It can be concluded that the sulfur cross-linking system was partially adsorbed onto the outer surface of the BDC filler, resulting in a slower curing process [12,13].

#### 4.2.2. Cross-Linking Density

Figure 3 shows the effect of BDC on the cross-linking density of the SBR composites. Increasing the concentration of the protein filler caused a gradual increase in the degree of cross-link density in the SBR vulcanizates. These results are in agreement with the previous rheometer measurements.

The higher degree of cross-link density in the SBR/BDC composites can be explained by the reactivity of groups originating in the BDC filler and the elastomeric matrix. The most probable interactions occur between the styryl group in rubber and the polar fragments of the collagen dust, as shown in Figure 4. The interactions between the BDC dust and the elastomer matrix were also confirmed by infrared analysis (Figure 5 and Figure 6).

#### 4.2.3. FTIR Analysis

In the next part of the study, the composites were subjected to FTIR analysis. Figure 5 and Figure 6 show the sample spectra of SBR composites containing various amounts of grinding dust, as well as the reference spectrum. Based on the BDC spectra of both SBR composites, a small band can be seen in the 3400 to 3250 cm^−1^ range (slightly larger for SBR20), which is associated with vibrations of the -OH group (Figure 5). In the range of 3100 to 3000 cm^−1^, a peak is visible from the stretching vibrations of the C–H bonds of the aromatic ring that form part of the SBR vulcanizate side groups [29]. In the 2950 to 2800 cm^−1^ range, intense peaks are visible from the vibrations of the C–H groups present in the material, both from the styrene aromatic ring and from the protein chain. The peak in the range of 1700 to 1630 cm^−1^ is caused by the presence of C=C diene groups. The visible change and shifts in absorption bands at 1750 to 1730 cm^−1^ can also be associated with the chromium complex and the free carboxyl group of oligomers that is generated by the fragmentation of peptide chains. The bands in the 1665 to 1550 cm^−1^ range are derived from the vibrations of the C=O peptide bond groups [13,14]. The deviation and significant decrease in the intensity of the absorption band at 1665 cm^−1^, as well as its shift to 1630 cm^−1^ in the SBR or SBR20 spectra relative to the spectrum for BDC, is caused by chromium, which can be coordinated with active collagen centers (amides I and II), thus forming a chromium complex [6]. Together with the shift, this significant reduction in the band (1730 cm^−1^) for the BDC spectrum compared to that for SBR20 supports our supposition that the probable mechanism of interaction between the elastomer macromolecule and BDC structure is as shown in Figure 4 [6].

The intensity of the BDC bands also changes depending on the amount of dust. In comparison with the spectrum for BDC alone (Figure 1 and Figure 6), the intense band at 1665 cm^−1^ derived from primary and secondary amides –(H_2_N)CO; –R(NH)CO significantly reduces its intensity, which may indicate the possibility of interactions with the C-C groups in the SBR aromatic rubber ring. The absorption band in the 1600 to 1550 cm^−1^ range is also caused by the presence of C–C bonds in the aromatic ring. The intense bands at 1546 cm^−1^ are derived from the stretching vibrations of the –CH_2_ and –CH_3_ groups. The small band around 1300 cm^−1^ is from H–O–H groups. It is associated with the presence of the small amount of water in zinc oxide. The band between 1018 and 1080 cm^−1^ is related to the absorption of stretching vibrations originating from C–C moieties in the aromatic ring, which are much more intense when BDC is applied to the SBR structure. This may indicate some interaction between the filler and the elastomeric matrix. The clear peak around 831 cm^−1^ is responsible for the presence of disubstituted alkenes. Intense vibrations are visible at 746 cm^−1^, which may be due to vibrations by hydroxylysine N–H groups present in the BDC or at 581 cm^−1^ from sulfur groups S–H or S–S [13,30].

#### 4.2.4. Mechanical and Hardness Tests

The effects of the BDC on the mechanical properties of the SBR vulcanizates are presented in Table 3. The incorporation of the protein filler enhanced the tensile strength (TS) of the SBR composites in almost all cases, except for sample SBR5 containing the lowest amount of BDC (5 phr). The application of BDC significantly increased the stiffness of the composites, as reflected in the higher stress at 300% elongation (SE300). This parameter is related to the cross-link density of the composites and confirms the results from the previous swelling measurements and rheometric studies. The highest values for tensile strength and SE300 modulus were noted when 10 phr of BDC was added. These values then decreased upon further addition of the protein filler. This may be related to the greater tendency of hydrophilic BDC filler to agglomerate at higher concentrations in the SBR matrix. After the addition of BDC, the elongation at break was significantly reduced in all of the BDC-based composites, most likely due to the increase in crosslink density. These observations are in agreement with rheometric studies and swelling experiments.

The incorporation of BDC in SBR improved the hardness of the prepared composites in comparison to the reference sample (Table 3). The hardness increased with increasing filler content. The most pronounced increase in hardness was observed in the vulcanizate containing 30 phr of BDC filler. The higher hardness values of the SBR/BDC samples can be explained by the presence of the BDC filler and the increased cross-link density of the composites. Compared with other studies [12,13,14] where buffing dust BDC was introduced into, e.g., a matrix of rubbers with special applications, namely, acrylonitrile butadiene rubber (NBR) and carboxylated acrylonitrile butadiene rubber (XNBR), there was an increase in the tensile strength and elongation at break. This could be related to the different ways in which the buffing dust was applied to the polymer matrices. In the previous studies, BDC was introduced by direct mixing with a cross-linking process activator, zinc oxide (ZnO).

#### 4.2.5. SEM Analysis

Filler dispersion is an important factor that affects the properties of rubber composites. Therefore, the next stage of the study investigated the distribution of BDC in the SBR matrix. Figure 7a–e presents SEM micrographs of cross sections of the SBR/BDC composites at different magnifications. The micrographs of SBR10 and SBR30 composites at 1.00× magnification show a rather homogeneous distribution of components in the elastomeric matrix. The reference sample showed fine, suspended particles of curing components, which are also visible in the photos of composites with the addition of BDC. The magnifications at 50.00 and 100.00 for BDC-based samples show irregular fragments of suspended particles, with diameters of approximately 100 or 200 nm. It appears that the agglomerates in the sample containing 10 phr of BDC were smaller in size than those in the SBR30 composite.

#### 4.2.6. DSC & TGA analysis

The results of the thermogravimetric analysis (TGA) and differential scanning calorimetry (DSC) of selected SBR and SBR10 composites are shown in Table 4, as well as in Figure 8 and Figure 9.

The introduction of the BDC filler to SBR did not significantly affect the temperature of thermal decomposition. In thermogravimetric analysis, a 5% weight loss was observed for the SBR10 composite at a slightly lower temperature than the reference sample. As can be seen from the standard deviation, the maximum decomposition temperatures of all the composites are similar. The total weight loss of the tested elastomers during decomposition was 95% for the reference sample and 90–91% for the composites filled with BDC.

The glass transition temperature for pure styrene–butadiene rubber is reported in the literature data as 48/65 °C [31]. The DSC revealed that the glass transition temperature Tg begins for the SBR composite at onset, i.e., at −46.6 °C, whereas for SBR10, Tg begins at −49.60 °C (Table 4, Figure 9). The glass transition temperature is shifted relative to the reference composition (SBR, −42.08 °C) towards lower values for the composition containing BDC dust (−45.45 °C). This shift is clear in the case of the sample with 10 parts wt. BDC. It suggests that BDC filler acts as a plasticizer in SBR compounds. The heat capacity of the SBR composite was 0.450 Jgˆ-1Kˆ-1, whereas for SBR10, heat capacity decreased to 0.399 Jgˆ-1Kˆ-1. 

#### 4.2.7. Swelling Balance in Water, Color Research, Soil Test, and Thermo-Oxidative Aging Process

There is considerable interest in accelerated aging processes and the behavior of polymer materials in contact with various factors. The dust-based elastomer composites were therefore subjected to water infraction into the material structure (Qw, -), soil tests, color change, and accelerated thermo-oxidative aging (ΔT, -). Figure 10 presents the results of equilibrium swelling in water. The addition of protein filler caused swelling extension relative to the unfilled sample. Due to the hydrophilic character of protein, higher water absorption was observed. The absorption of water facilitated the penetration of microorganisms, and thus the biodegradation process was more rapid.

The next test was to determine the surface color change (dE × ab, -) of the vulcanizates exposed to elevated temperature and biological properties (soil tests). Table 5 shows photographs taken before and after biodegradation. As can be seen, topographic changes occurred in the composites containing the bionic biofiller. There were discolorations in the material where micro- and macro-cracks formed on the surface. The change in the appearance was influenced by the action of the biological agents found in soil, which, by breaking down the protein filler, contributed to the deterioration of the properties of the tested vulcanizates.

Factor dL × ab was found to change the color of the sample after degradation relative to the sample before degradation. The highest values for dL × ab were found for the native composite, SBR, which proves that the other samples were protected against thermo-oxidative aging, especially in the case of the 10 or 20 phr biopolymer. Higher values for this indicator are associated with greater color deviation in the case of biodegradation. The action of soil organisms promotes the degradation of the filler, which leads to a loss of consistency in the material structure. This indicates much greater susceptibility to biodegradation in the case of vulcanizates containing the protein filler. The changes in the color of the composites were affected by the aging conditions, i.e., temperature, humidity, as well as microorganisms contained in soil in soil tests.

It was found that addition of protein filler resulted in acceleration of biodegradation in almost all of the samples. The mechanical deterioration of composites with BDC dust to the soli test can also be seen on the basis of the aging coefficient (ΔT, -), as the combination of tensile strength and elasticity before and after aging (Table 6).

On the basis of soil tests, it appears that biodegradation caused a decrease in tensile strength (Table 5 and Table 6). This decrease is probably due to filler degradation, which causes a reduction in intermolecular bonding [32]. It was found that the addition of the protein filler resulted in accelerated biodegradation in almost all the samples. The hardness of samples was greater after the soil tests than before the tests. The greater hardness was a consequence of the biodegradation process. Increased hardness was noticed as a result of thermo-oxidative aging relative to the non-aged samples. The noticeable increase in hardness may be associated with greater cross-link density, secondary cross-linking, or the evaporation of water derived from the hydrophilic filler. Increasing the filler load resulted in higher hardness values.

## 5. Conclusions

In this study, collagen grinding dust was applied as a biodegradable filler in the SBR polymer, to study its effects on the mechanical properties, biodegradation, and thermal stability of the SBR vulcanizates. Due to its particle size of less than 1 micrometer, BDC has potential to be used in SBR rubber as nanofillers. The elongated rod-like shape promotes more favorable dispersion of the filler in the elastomer medium, as shown by SEM photos. The presence of a scleroprotein additive results in a higher degree of cross-linking in the case of vulcanizates, which in turn increases the stiffness and hardness of the composites. Interactions may occur between the components of the elastomer matrix, as evidenced by the increased intensity of IR absorption bands in the range of 1000 to 1750 cm^−1^. A greater weight share of filler results in shift of glass transition temperatures towards lower values, and improved hardness. As a result of the biological or thermal aging with oxygen, greater water infiltration was observed in the structures of the material produced on the basis of SBR*30* rubber. Surface color stabilization was also visible, e.g., under the influence of ultraviolet radiation. The introduction of BDC dust had a stabilizing effect on thermo-oxidative aging processes, due to the antioxidant properties of collagen dust itself. The obtained studies demonstrated that sifted and thermostabilized grinding dust can play a role as a biodegradable filler in polymer materials and, due to intense black color, can act as coloring additive. As expected, the thermal stability of the composites increased, most likely due to the incorporation of chromium ions in the polymer network. The results of this study expand the possibilities for managing hazardous tanning wastes in the form of dust from chrome tanned leathers, while reducing their environmental impact.

## Figures and Tables

**Figure 1 materials-13-01498-f001:**
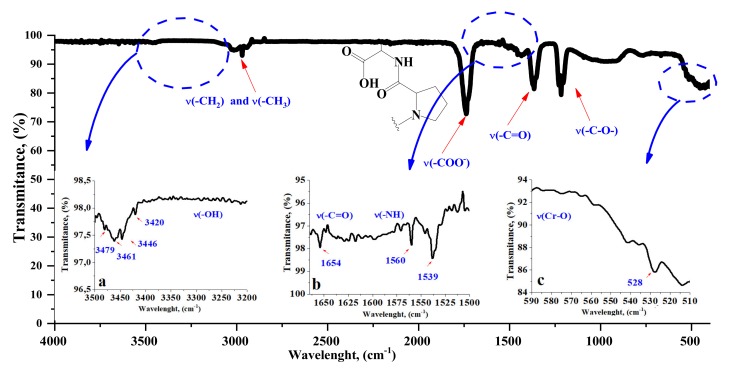
IR spectra of buffing dusts of collagen with ranges: (**a**) 3500–3200 cm^−1^; (**b**) 1650–1500 cm^−1^; (**c**) 590–510 cm^−1^.

**Figure 2 materials-13-01498-f002:**
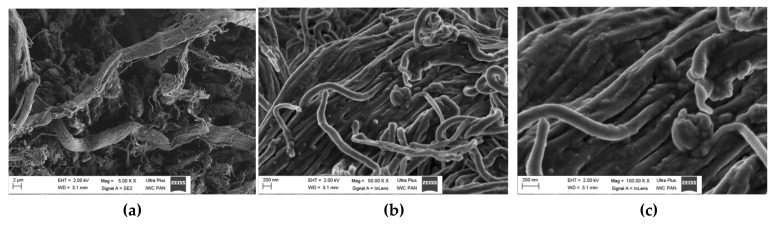
SEM images of buffing dust collagen (BDC): (**a**) 5000× magnification, (**b**) 50,000× magnification, and (**c**) 100,000× magnification.

**Figure 3 materials-13-01498-f003:**
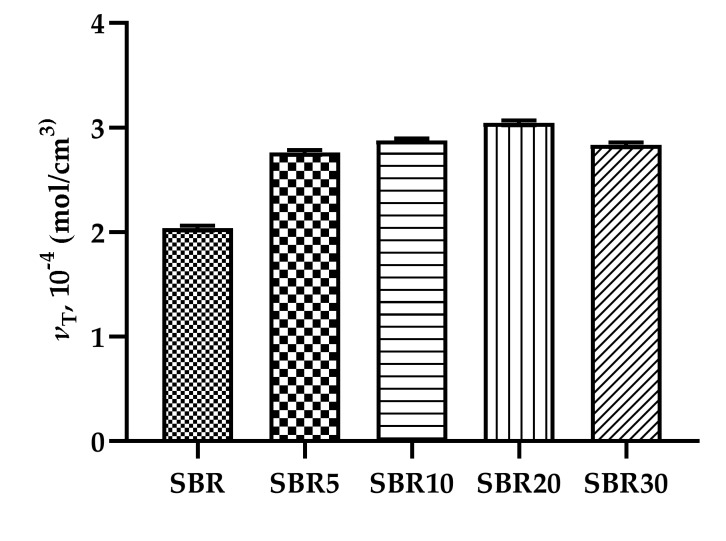
Influence of BDC on the cross-linking density of SBR vulcanizates.

**Figure 4 materials-13-01498-f004:**
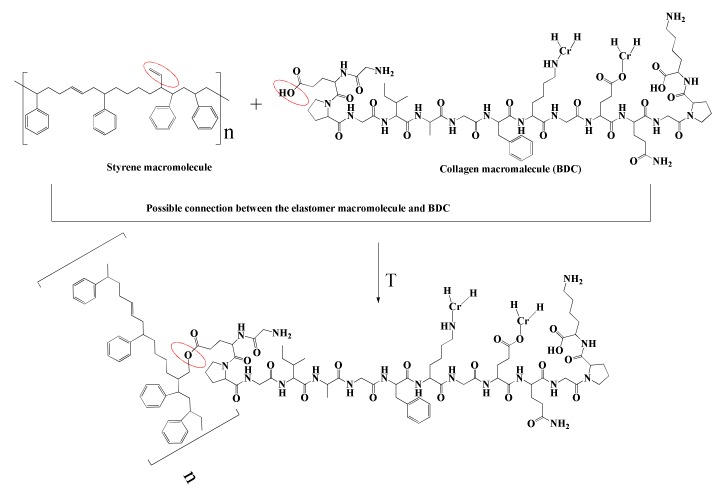
Possible mechanism of interaction between the elastomer macromolecule and BDC structure.

**Figure 5 materials-13-01498-f005:**
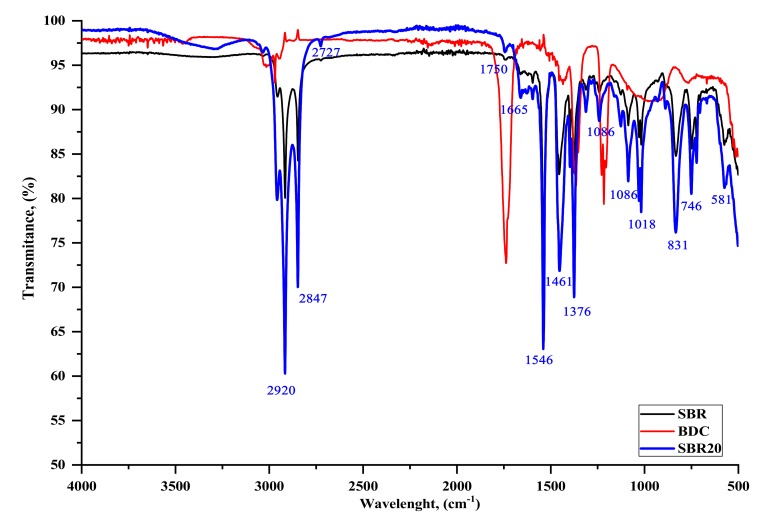
Fourier transform infrared spectroscopy (FTIR) spectrum of samples containing 0 and 20 phr BDC in SBR composites relative to the BDC spectrum.

**Figure 6 materials-13-01498-f006:**
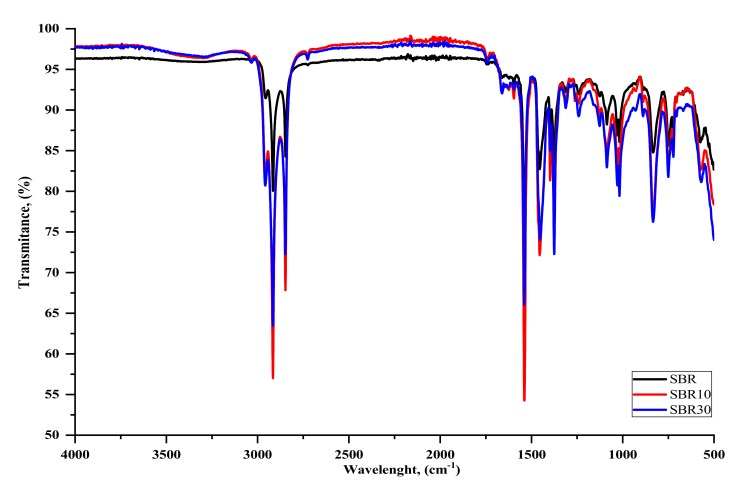
Comparison of the FTIR spectra for the standard sample (SBR), 10, and 30 phr BDC in SBR composites.

**Figure 7 materials-13-01498-f007:**
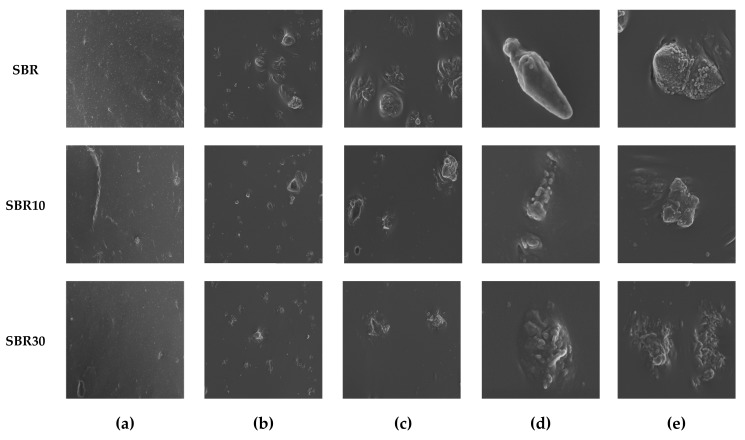
SEM images of SBR, SBR10, and SBR30 composites at different magnifications: (**a**) 1.00×, (**b**) 10.00×, (**c**) 25.00×, (**d**) 50.00×, and (**e**) 100.00×.

**Figure 8 materials-13-01498-f008:**
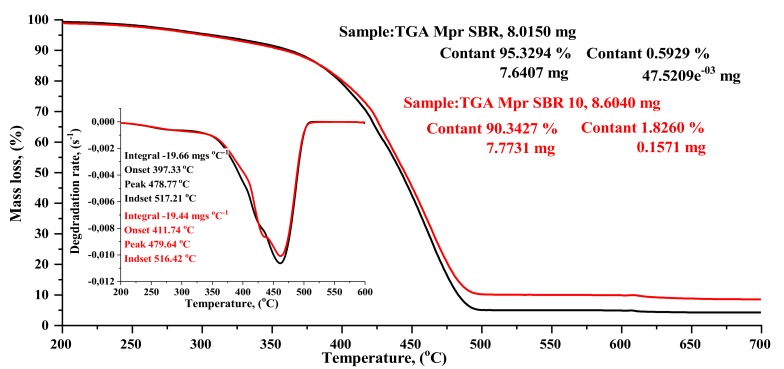
Thermogravimetric analysis (TGA) curves of SBR composites: reference sample SBR (black curve) and SBR10 sample (red curve).

**Figure 9 materials-13-01498-f009:**
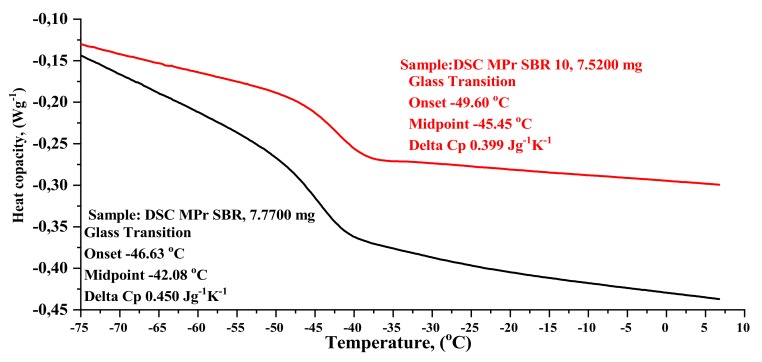
Comparative analysis of DSC results: SBR—unfilled (black curie) and SBR10 (10 phr of BDC—red curve).

**Figure 10 materials-13-01498-f010:**
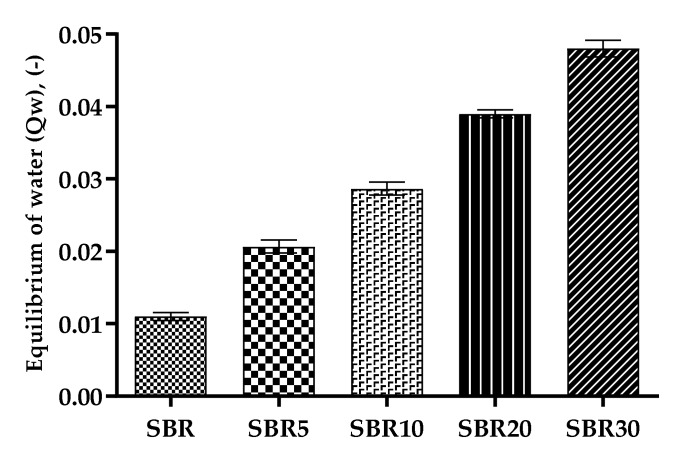
Equilibrium of swelling in water results from SBR vulcanizates.

**Table 1 materials-13-01498-t001:** Composition of the Styrene–Butadiene Rubber (SBR) composites.

Symbol	SBR	SBR5	SBR10	SBR20	SBR30
SBR (phr)	100	100	100	100	100
BDC (phr)	0	5	10	20	30
ZnO (phr)	5	5	5	5	5
Sulphur (phr)	2.5	2.5	2.5	2.5	2.5
MBTS (phr)	1.5	1.5	1.5	1.5	1.5
Stearic acid (phr)	2	2	2	2	2

Vulcanizates were prepared in hydraulic presses. Samples in the form of 100 × 50 mm films were prepared by pressing under 150 MPa at 150 °C.

**Table 2 materials-13-01498-t002:** Influence of BDC on the rheometric properties of SBR compounds.

Symbol	L_L_ (dNm)	∆L (dNm)	τ_02_ (min)	τ_90_ (min)
SBR	11.0 ± 0.3	39.9 ± 1.1	1.2 ± 0.1	24.5 ± 1.3
SBR5	13.9 ± 0.1	54.8 ± 1.9	2.9 ± 0.1	30.0 ± 1.4
SBR10	12.0 ± 0.2	61.6 ± 2.2	2.0 ± 0.1	33.0 ± 1.1
SBR20	13.0 ± 0.1	51.7 ± 1.9	2.9 ± 0.2	37.0 ± 1.0
SBR30	14.0 ± 0.1	47.5 ± 1.4	3.5 ± 0.2	38.5 ± 1.2

L_L_—minimum torque moment (dNm); ∆L—the decrease of torque moment (dNm) (∆L = L_HR_ − L_L_); τ_02_—scorch time (min); τ_90_—time of vulcanization (min).

**Table 3 materials-13-01498-t003:** Influence of BDC filler on the mechanical properties of SBR composites.

Parameter	SBR	SBR5	SBR10	SBR20	SBR30
TS (MPa)	3.34 ± 0.1	3.73 ± 0.2	4.78 ± 0.3	4.91 ± 0.3	3.94 ± 0.2
SE300 (%)	2.15 ± 0.5	2.41 ± 0.1	2.60 ± 0.2	3.07 ± 0.6	3.70 ± 0.1
E_b_ (%)	541 ± 4.2	228 ± 6.1	254 ± 5.5	322 ± 7.8	267 ± 4.9
H (°Sh)	44.08 ± 0.9	48.22 ± 1.2	49.20 ± 1.1	50.85 ± 0.8	53.88 ± 0.7

**Table 4 materials-13-01498-t004:** Thermal characteristics of SBR and SBR10.

Symbol	T_5%_ (°C)	T _peak (DTG)_ (°C)	Δm _total_ (%)	T_g_ (°C)
SBR	305	478.77	95.32	−42.08
SBR10	299	479.64	90.34	−45.45

T_5%_—decomposition temperature at 5% mass loss, T_p(DTG)_—temperature of maximum conversion rate on the DTG curve, Δm _total_—total mass loss during thermal decomposition, T_g_—glass transition temperature (standard deviations: T_5_, T_p (DTG) ± 2 °C_; Δm _total ± 0.6%; Tg ± 2°C_)

**Table 5 materials-13-01498-t005:** Topography photos of surface changes of SBR composites before and after biodegradation tests.

Composite Symbol	SBR	SBR5	SBR10	SBR20	SBR30
**BEFORE BIODEGRADATION**	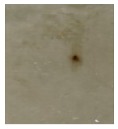	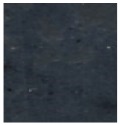	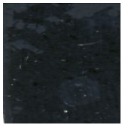	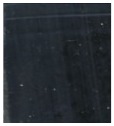	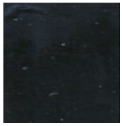
**dL × ab**	3.84	0.99	0.20	0.26	0.46
**AFTER BIODEGRADATION**	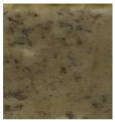	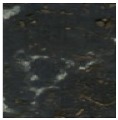	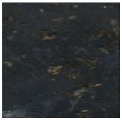	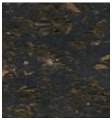	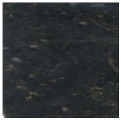
**dL × ab**	7.35	2.63	1.34	4.52	3.18

**Table 6 materials-13-01498-t006:** Mechanical properties after biodegradation.

Effects of Aging and Biological on the Properties of SBR Composites					
**Composite symbol**	**SBR**	**SBR5**	**SBR10**	**SBR20**	**SBR30**
**Aging coefficient ∆T(-)**					
Thermal aging	0.93 ± 0.04	1.42 ± 0.02	0.84 ± 0.03	1.14 ± 0.01	0.92 ± 0.03
Soil test	0.94 ± 0.01	0.83 ± 0.01	0.67 ± 0.02	0.56 ± 0.01	0.51 ± 0.02
**Hardness of obtained composites H (°Sh)**					
Thermal aging	45.17 ± 0.12	51.95 ± 0.08	53.33 ± 0.14	54.25 ± 0.23	58.15 ± 0.18
Soil test	45.00 ± 0.20	46.90 ± 0.13	48.07 ± 0.21	49.27 ± 0.10	53.03 ± 0.09

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
