# Peer review of "Hazardous Waste Management of Buffing Dust Collagen"

_materials, 2020, doi:10.3390/ma13071498_

Round 1
Reviewer 1 Report
The paper needs some changes as major revision.
Some important results should be added to the abstract.
It is better to delete the Figure 1.
The methodology should be cited by proper references.
The discussion needs more comparison with published studies.
Lines 62-65 should be moved to the conclusion section.
Fonts of some of the figures are too big.
If SBR 30 gave the best results, what about SBR 40 or SBR 50? it is good to know the trend of the results after SBR 30.
The reference list should be expanded to more recent studies (2017-2020).
Author Response
Response to Reviewer 1:
Q1: The paper needs some changes as major revision.
A1: The article has undergone general changes.
Q2: Some important results should be added to the abstract.
A2: Summary changed.
Q3: It is better to delete the Figure 1.
A3: I deleted drawing 1.
Q4: The methodology should be cited by proper references.
A4: The citations in the methodology have changed.
Q5: The discussion needs more comparison with published studies.
A5: The discussion of the results was subject to changes and a more thorough analysis in accordance with the cited literature.
Q6: Lines 62-65 should be moved to the conclusion section.
A6: Verses 62-65 have been moved to the conclusions.
Q7: Fonts of some of the figures are too big.
A7: Fonts have been changed to smaller one.
Q8; If SBR 30 gave the best results, what about SBR 40 or SBR 50? it is good to know the trend of the results after SBR 30.
A8: The tests performed present data on the composition of the committees with the share of the additive tested in quantities of 5 to 30 parts by weight. The study of the impact of participation in the amount of 40 and 50 parts by weight is the subject of further research and will be included in the another scientific dissertation.
Q10: The reference list should be expanded to more recent studies (2017-2020).
A10: The literature has been expanded with new reports from 2017-2020.

Reviewer 2 Report
SBR was filled into BDC to obtaine a biodegradable SBR-BDC composites, which is interesting could be accepted by the journal after some major revisions:
(1) English needs to be polished. For exmple, in "The leather industry is one of the most polluting", the most polluting what?
(2) The order needs to be reversed: "2.1.1. Buffing dust collagen (BDC)" and "2.1.2. Other ingredients". Besides, they should be modified.
(3) If an abbreviation was given for a full name, it should be used thought the following manuscript. For example, buffing dust collagen (BDC).
(4) The purity of the reagents shoud be given.
(5) The conditon parameters should be optimized.
(6)The low intensity band at 1410 cm-1 corresponding to the valence vibrations of the carbonyl group (C = O) in the carboxylate ion (-COO-) is also more visible in the dust spectrum? FT-IR analyses are wrong!
(7) EDS elemental mapping should be provided.
(8) A comparison with previously reported results should be given.
Author Response
Response to Reviewer 2:
(Q1) English needs to be polished. For exmple, in "The leather industry is one of the most polluting", the most polluting what?
(A1): Yes you are right - it has been corrected.
(Q2) The order needs to be reversed: "2.1.1. Buffing dust collagen (BDC)" and "2.1.2. Other ingredients". Besides, they should be modified.
(A2): I changed the order of these chapters and modified them.
(Q3) If an abbreviation was given for a full name, it should be used thought the following manuscript. For example, buffing dust collagen (BDC).
(A3): So I tried to use the same BDC abbreviation and corrected wherever it did not appear
(Q4) The purity of the reagents should be given.
(A4): I added the purity of the reagents.
(Q5) The condition parameters should be optimized.
(A5): Condition parameters have been optimized
(Q6)The low intensity band at 1410 cm-1 corresponding to the valence vibrations of the carbonyl group (C = O) in the carboxylate ion (-COO-) is also more visible in the dust spectrum? FT-IR analyses are wrong!
(A6): I agree, the sentence has been deleted, changes have been made to the FTIR analysis descriptions
(Q7) EDS elemental mapping should be provided.
(A7): Mapping has not been carried out for this batch of materials, it will be carried out for warehouses with increased BDC content.
(Q8) A comparison with previously reported results should be given.
(A8): A comparison with previous results has been made.

Reviewer 3 Report
Dear Authors,
The manuscript is very interesting and is well written in good English and in an adequate format for Material. The use of the waste and the development of green processes is a very good presented and the conclusion supported by results. The manuscript can be accept in present form.
Author Response
Response to Reviewer 3:
Thank you very much for your time reviewing the article. So far, I have not met with such a friendly review. Once again, thank you very much.
Authors

Round 2
Reviewer 1 Report
The paper is improved as the reviewers requested. While minor correction is needed before publication.
The paper needs proofread by a native speaker. Some sentences are difficult to understand.
Lines 136 to 160 needs to be cited.
The fonts of numbers in figure 5 should be smaller.
The authors said" The tests performed present data on the composition of the committees with the share of the additive tested in quantities of 5 to 30 parts by weight. It is good to add the logic of why the authors used this range in the paper.
References are acceptable in this form. Just check why some references have DOI and some not.
Reference number 25 is incomplete.
Round 2
Response for Reviewer 1
The paper is improved as the reviewers requested. While minor correction is needed before publication.
Q1. The paper needs proofread by a native speaker. Some sentences are difficult to understand.
A1. Thank you for the review. All changes are marked in green. The language has been corrected with the help of a native speaker
Q2. Lines 136 to 160 needs to be cited.
A2. The citations have been added.
Q3. The fonts of numbers in figure 5 should be smaller.
A3. The font size has been changed in both Figures 5 and 6 and also Fig. 1.
Q4. The authors said" The tests performed present data on the composition of the committees with the share of the additive tested in quantities of 5 to 30 parts by weight. It is good to add the logic of why the authors used this range in the paper.
A4. Fillers introduced into elastomers can have various functions. Once they were only required to perform a filling function, now there are also used to improve thermal and electrical conductivity, reduce oxidation and friction, increase strength and hardness, etc.
The BDC dust used belongs to the group of reinforcing fillers, which are introduced in quantities greater than 5 parts by weight. Multiples of these weights are most often chosen, which is why 5, 10, and 30 parts by weight were chosen. The filling effect of BDC reduces the cost of producing the composite. However, no more than 40 or 50 parts by weight may be introduced, because more than this quantity worsens most of the physico-chemical parameters.
Q5. References are acceptable in this form. Just check why some references have DOI and some not.
A5. The same format has been used for all References
Q6. Reference number 25 is incomplete.
A6. It has been corrected.

Reviewer 2 Report
It could be accepted.
Round 2
Response for Reviewer 2
Thank you for the review. All changes are marked in red. The language has been corrected with the help of a native speaker.
